# Effects of a Novel GPR55 Antagonist on the Arachidonic Acid Cascade in LPS-Activated Primary Microglial Cells

**DOI:** 10.3390/ijms22052503

**Published:** 2021-03-02

**Authors:** Soraya Wilke Saliba, Franziska Gläser, Anke Deckers, Albrecht Keil, Thomas Hurrle, Matthias Apweiler, Florian Ferver, Nicole Volz, Dominique Endres, Stefan Bräse, Bernd L. Fiebich

**Affiliations:** 1Neurochemistry and Neuroimmunology Research Group, Department of Psychiatry and Psychotherapy, Medical Center-University of Freiburg, Faculty of Medicine, University of Freiburg, D-79104 Freiburg, Germany; wilkesaliba@yahoo.com.br (S.W.S.); albrecht.keil86@gmail.com (A.K.); matthias.apweiler@uniklinik-freiburg.de (M.A.); florianferver@gmail.com (F.F.); 2Institute of Organic Chemistry, Karlsruhe Institute of Technology (KIT), D-76131 Karlsruhe, Germany; glaeser.f@web.de (F.G.); hurrle.thomas@gmail.com (T.H.); nicole.volz@hotmail.com (N.V.); braese@kit.edu (S.B.); 3Institute of Biological and Chemical Systems-Functional Molecular Systems (IBCS-FMS), Karlsruhe Institute of Technology (KIT), Hermann-von-Helmholtz-Platz 1, D-76344 Eggenstein-Leopoldshafen, Germany; anke.deckers@kit.edu; 4Section for Experimental Neuropsychiatry, Department of Psychiatry and Psychotherapy, Medical Cente-University of Freiburg, Faculty of Medicine, University of Freiburg, D-79104 Freiburg, Germany; dominique.endres@uniklinik-freiburg.de

**Keywords:** GPR55, prostaglandin E_2_, cyclooxygenase, microglia, neuroinflammation

## Abstract

Neuroinflammation is a crucial process to maintain homeostasis in the central nervous system (CNS). However, chronic neuroinflammation is detrimental, and it is described in the pathogenesis of CNS disorders, including Alzheimer’s disease (AD) and depression. This process is characterized by the activation of immune cells, mainly microglia. The role of the orphan G-protein-coupled receptor 55 (GPR55) in inflammation has been reported in different models. However, its role in neuroinflammation in respect to the arachidonic acid (AA) cascade in activated microglia is still lacking of comprehension. Therefore, we synthesized a novel GPR55 antagonist (**KIT 10**, 0.1–25 µM) and tested its effects on the AA cascade in lipopolysaccharide (LPS, 10 ng / mL)-treated primary rat microglia using Western blot and EIAs. We show here that **KIT 10** potently prevented the release of prostaglandin E_2_ (PGE_2_), reduced microsomal PGE_2_ synthase (mPGES-1) and cyclooxygenase-2 (COX-2) synthesis, and inhibited the phosphorylation of Ikappa B-alpha (IκB-α), a crucial upstream step of the inflammation-related nuclear factor-kappaB (NF-κB) signaling pathway. However, no effects were observed on COX-1 and -2 activities and mitogen-activated kinases (MAPK). In summary, the novel GPR55 receptor antagonist **KIT 10** reduces neuroinflammatory parameters in microglia by inhibiting the COX-2/PGE_2_ pathway. Further experiments are necessary to better elucidate its effects and mechanisms. Nevertheless, the modulation of inflammation by GPR55 might be a new therapeutic option to treat CNS disorders with a neuroinflammatory background such as AD or depression.

## 1. Introduction

Neuroinflammation is considered a double-edged sword, in which the acute response is defined as a protective event and, on the other hand, chronic neuroinflammation is detrimental [1]. Several studies demonstrated the contribution of the inflammatory process to the pathophysiology of the different neurological and neuropsychiatric disorders such as Alzheimer’s disease (AD), Parkinson’s disease (PD), schizophrenia, and depression. Neuroinflammation is well characterized by the activation of glial cells, mainly microglia, which leads to the release of inflammatory mediators [2]. Microglial cells are the first line of defense in the central nervous system (CNS) and exert an anti-inflammatory role being neuroprotective and implicated in the resolution of the inflammatory event by phagocytosis and tissue repair. However, for reasons that are still not clear, microglia have also a detrimental role, where the deregulated microglia response by inducing a pro-inflammatory phenotype releasing pro-inflammatory mediators including cytokines such as tumor necrosis factor (TNF) α, interleukin-6 (IL-6), and IL-1β, prostaglandins (PGs), reactive oxygen species (ROS), and nitric oxide (NO) resulting in progressive neurotoxic consequences and neurodegeneration [3,4,5].

The G-protein-coupled receptor, GPR55, is highly expressed in the CNS [6], but also peripheral tissues [7]. Moreover, our group and others have demonstrated the expression of GPR55 in microglial cells [8,9]. The role of GPR55 in inflammation has been reported in in vitro and in vivo models [10,11,12] and its mediated effects remain controversial and depended on the cell type and stimulation. The activation of GPR55 by its agonist O-1602 increased pro-inflammatory cytokines and cell cytotoxicity in monocytes and natural killer (NK) cells stimulated with lipopolysaccharide (LPS) [11]. Corroborating with this, an antagonist of GPR55, CID16020046, as well as GPR55^−/−^ knockout mice decreased pro-inflammatory cytokines in mice models of colitis comparable to human inflammatory bowel disease (IBD) [12]. In hyperalgesia associated with inflammatory and neuropathic pain, increased anti-inflammatory cytokines (IL-4 and IL-10) and also pro-inflammatory interferon (IFN) were observed in GPR55^−/−^ knockout mice [10]. On the other hand, treatment of IL-1β stimulated neuronal cells with O-1602 and chronic systemic inflammation induced by LPS administration in mice demonstrated neuroprotection and an anti-inflammatory profile [13].

GPR55 couples to Gα12/13 proteins which leads to the activation of ras homolog gene family members A (RhoA) and Rho-associated protein kinase (ROCK) triggering the activation of the phospholipase C pathway increasing the intracellular Ca^2+^ and phosphorylation of extracellular signal-regulated kinase (ERK) and p38 MAPK [14,15]. The MAPK signaling is involved in the regulation of inflammatory events [16] and we have demonstrated in activated microglia that the activation of JNK, ERK, and p38 MAPK are implicated in the regulation of COX-2/mPGES-1/PGE_2_ levels, important proteins involved in neuroinflammation also mediating fever [17,18,19]. We demonstrated that GPR55 may regulate COX-2/mPGES-1/PGE_2_ in several steps.

GPR55 is endogenously activated by lysophosphatidylinositol (LPI, lysoPI), a bioactive lipid involved in a range of pathologies such as obesity, cancer, and inflammation [20,21], and by different cannabinoids [22]. LPI has been demonstrated to reveal anti-inflammatory effects in microglial cells by decreasing cytokine release and suppressing intracellular generation of reactive oxygen species [23]. Furthermore, a neuroprotective effect in an ex-vivo model of excitotoxicity, mediated by microglial cells, has been shown [24]. 

These studies demonstrated the involvement of GPR55 in different aspects of inflammation, suggesting this receptor to be an interesting target to modulate neuroinflammation by interfering with the arachidonic acid (AA) cascade. In the present study, a novel GPR55 antagonist based on the coumarin structure (**KIT 10**) was synthesized, and the effects of **KIT 10** on the production of PGE_2_ induced by LPS in primary microglial cells was investigated. We have observed the anti-neuroinflammatory effect of **KIT 10** by potently inhibiting LPS-induced PGE_2_ synthesis in microglia and this effect was associated with the reduction of mPGES-1 and COX-2 protein synthesis, but not through the inhibition of COX-1 and COX-2 enzyme activity. Interestingly, **KIT 10** affected the NF-κB pathway, whereas MAPK activation was not affected.

## 2. Results

### 2.1. Chemical Part

#### Chemical Synthesis

The synthesis of **KIT 10** (7-(1-butylcyclohexyl)-3-(4-fluorobenzyl)-5-hydroxy-2*H*-chromen-2-one, Scheme 1) was achieved according to a straightforward reaction protocol developed by our group [25,26,27,28]. In the first step, an N-heterocyclic carbene-catalyzed (NHC) reaction between salicylic aldehyde **1** and cinnamic aldehyde **2** yields coumarin **3** in 25% yield. The precursor of the NHC is the shown imidazolium phosphate. Coumarin **3** was then derivatized in a subsequent reaction step leading to coumarin **4**, where a deprotection with boron tribromide lead to a coumarin (named **KIT 10** throughout the manuscript) in 87% yield. This compound revealed antagonistic activities (for the method see [28]) on GPR55 with an IC_50_ of 3.91± 1.03 µM (author’s unpublished data).

### 2.2. Biological Part

#### 2.2.1. Effects of **KIT 10** on Cell Viability

We first evaluated the effects of **KIT 10** on cell viability in primary rat microglia using an ATP assay to exclude possible biological inhibitory effects due to a reduction of cell viability. As shown in Figure 1, **KIT 10** in all concentrations, as well as LPS, did not change cell viability compared to vehicle only (0.1% DMSO)-treated cells. DMSO (10%) was used as a cell toxic control.

#### 2.2.2. **KIT 10** Prevents LPS-Induced PGE_2_ Release Primary Microglial Cells

We next studied the effects of **KIT 10** on LPS-induced PGE_2_ in microglia. We observed a concentration-dependent inhibition of LPS-mediated PGE_2_ release starting with 1 µM and maximal effects using 10 and 25 µM with a calculated IC_50_ of 2.5 ± 0.93 µM (Figure 2). Since 10 µM was sufficient to potently prevent LPS-induced PGE_2_ synthesis, we used this concentration as a maximal concentration in further experiments.

#### 2.2.3. **KIT 10** Inhibits LPS-Mediated COX-2 and mPGES-1 Protein Synthesis in Primary Microglial Cells

Under an inflammatory stimulus, the enzymes COX-2 and mPGES-1 are responsible to synthesize prostaglandins [29]. As shown in Figure 3, LPS potently induced protein synthesis of COX-2 and mPGES-1. We show here that LPS-induced mPGES-1 synthesis was significantly inhibited by **KIT 10** in the concentrations of 5 and 10 µM, which decreased mPGES-1 levels about approx. 50% if compared to LPS control (Figure 3A). LPS-induced COX-2 protein synthesis (Figure 3B) was prevented only the highest concentration of 10 µM (approx. 40% inhibition). 

#### 2.2.4. **KIT 10** Didn’t Affect the COX Enzyme Activities

To verify that **KIT 10** mediated inhibition of PGE_2_ release is additionally due to direct suppression of COX enzymatic activities, we performed a COX enzyme activity assay. As shown in Figure 4, COX-1 and -2 activities were not affected by all concentrations of **KIT 10**. In contrast, COX enzyme activities were strongly inhibited by classical inhibitors of COX-1 (SC560, Figure 4A) and COX-2 (diclofenac sodium, Figure 4B).

#### 2.2.5. Effects of **KIT 10** on IκB-α Phosphorylation in LPS-Stimulated Primary Microglial Cells

Multiple evidence indicates that the phosphorylation and degradation of Ikappa B-alpha (IκB-α), as well as the phosphorylation of IKK-alpha/beta, are primarily responsible for the activation of the NF-kappaB (NF-κB) signaling pathway [7]. Therefore, we studied the effects of **KIT 10** on IκB-α phosphorylation, a crucial upstream step of the inflammation-related NF-κB signaling pathway. As shown in Figure 5, LPS-induces prominent phosphorylation of IκB-α. Treatment of microglial cells with increasing concentrations of **KIT 10** reversed LPS-induced IκB-α phosphorylation in the concentrations of 5 and 10 µM. 

#### 2.2.6. Effects of **KIT 10** on Mitogen-Activated Kinases (MAP Kinases): JNK, ERK 1/2 and p38 MAPK in LPS-Stimulated Primary Microglial Cells

MAPK signaling is involved in the regulation of inflammatory events [30] and we have demonstrated that the activation of JNK, ERK, and p38 MAPK are implicated in the regulation of COX-2/PGE_2_ levels in activated microglia [17,18,19]. We, therefore, explored the effects of **KIT 10** on MAPK signaling pathways in LPS-stimulated primary microglial cells. However, none of the LPS-phosphorylated and thus activated MAPKs [JNK (Figure 6A), ERK 1/2 (Figure 6B), p38 MAPK (Figure 6C)] were significantly affected by **KIT 10** although a tendency of weak inhibition was overserved for p38 MAPK and JNK

## 3. Discussion

Neuroinflammation is an important process to maintain homeostasis in the CNS. It is involved in the response against different invaders and deregulators such as pathogens, toxins, infections, and the accumulation of aggregated or modified proteins [31]. 

The neuroinflammatory response is mediated by glial cells, especially microglia. Microglial cells are the resident macrophages of the CNS, responsible for the initiation, amplification, and/or equalization of the inflammatory response by the synthesis of inflammatory mediators, like cytokines, prostaglandins (PGs), and free radicals [3,4,5]. 

In the present study, we demonstrated the anti-neuroinflammatory effects of a novel GPR55 antagonist, **KIT 10**, in LPS-activated primary microglial cells by potently inhibiting LPS-induced PGE_2_ synthesis. This effect was associated with the reduction of protein synthesis of mPGES-1 and COX-2, and not through the inhibition of COX-1/2 enzyme activity. Interestingly, **KIT 10** affected the NF-κB pathway, but not MAPKs.

With the intention to understand the effects of the novel GPR55 antagonist **KIT 10** in neuroinflammation, especially in microglia, we first eliminated the possibility that any effects observed may be caused by toxic effects of the compound. Therefore, we evaluated its effects on cell viability and observed that the compound is not toxic at all concentrations tested. 

We next investigated the effects of **KIT 10** on LPS-induced PGE_2_ release. Microglial cells are the most important source of PGE_2_ in neuroinflammation. The stimulation of primary microglia with LPS strongly increases PGE_2_ and its synthesizing enzymes COX-2 and mPGES-1 [32,33,34]. We have previously shown that GPR antagonists are inhibiting LPS-induced COX-2 and mPGES-1 protein levels and PGE_2_ synthesis in primary microglia [8]. Corroborating with our findings, the GPR55 antagonist CID16020046 was demonstrated to reduce COX-2 mRNA and protein levels and decreased the release of PGE_2_ of advanced glycation end products (AGEs) induced chondrocytes activation in chondrogenic cell line ATDC5 [35]. In mice models of intestinal inflammation (chronic drinking of 2.5% dextran sulfate sodium or by a single intrarectal application of trinitrobenzene sulfonic acid in C57BL/6 mice), chronic treatment with CID16020046 (20 mg/kg) significantly decreased COX-2 levels [12]. Furthermore, GPR55 knockout mice (GPR55^-/-^) revealed a decrease in COX-2 levels in a mouse model of colorectal cancer, using azoxymethane and dextran sulfate sodium [36].

We further investigated if the reduction of PGE_2_ levels may be mediated by a decrease of the enzymatic activity of COX, the mechanism of action of most nonsteroidal anti-inflammatory drugs (NSAIDs). Two COX isoforms exist, COX-1 and COX-2. COX-1 is constitutively expressed in almost every cell and under unstimulated conditions; primary microglial cells only express the COX-1 isoform [37]. The other isoform COX-2 is induced by cytokines and endotoxins but is also constitutive in some other cells such as neurons, endothelial, or smooth muscle cells [38,39]. In our previous study [8], we have demonstrated that GPR55 antagonists did not affect COX-2 activity, but increased COX-1 activity, being a beneficial property as COX-1 is responsible for gastric mucosa protection [40,41]. Here, we showed that **KIT 10** did not affect both COX activities, demonstrating that the reduction on the PGE_2_ levels by **KIT 10** is not due to direct inhibition of COX enzyme activity.

The activity of the transcription factor Nuclear factor-kappa B (NF-κB) is increased in acute CNS diseases such as stroke and severe epileptic seizures, as well as in chronic neurodegenerative conditions, including AD, PD, and amyotrophic lateral sclerosis [42]. Moreover, several studies indicate that degradation of IκBα, as well as the phosphorylation of IKKα/β, are primarily responsible for the activation of the NF-κB signaling pathway [43,44,45]. The endogenous ligand of GPR55, LPI, has been shown to activate NF-κB and nuclear factors of activated T-cells (NFAT). On the other hand, the GPR55 antagonist CID16020046 concentration-dependently inhibited the LPI effects on these transcription factors [46]. In GPR55 knockout mice (GPR55^-/-^), NF-κB levels are decreased in a mouse model of colorectal cancer [36]. In addition, the GPR55 antagonist CID16020046 showed inhibition on IκBα phosphorylation on AGEs induced activation of the chondrogenic cell line ATDC5 [35]. Further studies have to be performed to elucidate the upstream step on which **KIT10** exactly prevents the phosphorylation of IκBα. Besides interfering with the activation of IL-1R-associated kinase (IRAK) 1/2 and IKK, GPR55 antagonists might directly act at recruited adapters of TLR4 such as myeloid differentiation factor (MyD) 88 and TNF-receptor-associated factor (TRAF6). 

MAPK signaling is involved in the regulation of inflammatory events [16] and we have demonstrated that the activation of JNK, ERK, and p38 MAPK are implicated in the regulation of COX-2/mPGES-1/PGE2 levels in activated microglia [17,18,19]. Several studies have demonstrated that the activation of GPR55 induced the phosphorylation of ERK 1/2 [9,20,47,48] and p38 MAPK [30]. However, LPS-induced phosphorylation of JNK, p38 MAPK, and ERK was not affected by **KIT 10**.

In summary, we provided evidence that **KIT 10** interferes in several steps of the synthesis of PGE_2_ in LPS-activated microglia. This study provides significant insights on the potential anti-inflammatory activity of this GPR55 antagonist and described a novel compound that modulates prostaglandin production by microglia.

## 4. Materials and Methods 

### 4.1. Chemistry

We have synthesized a novel GPR55 antagonist inspired by tetrahydrocannabinol, which we have called **KIT 10**. It differs from the compound **KIT 17** (3-benzyl-6-hydroxy-5,7,8-trimethyl-2*H*-chromen-2-one) used in our previous study [8], which is a small generic coumarin and structurally different from **KIT 10** (7-(1-butylcyclohexyl)-3-(4-fluorobenzyl)-5-hydroxy-2*H*-chromen-2-one). The position of the hydroxy groups (*meta/para*) causes a different acidity and thus different binding to proteins. Also, the bulky cyclohexyl ring in **KIT 10** increases the hydrophilic character of the western portion of this compound. Thus, the compounds are very different from each other.

Commercially available reagents and solvents for the synthesis were obtained from commercial suppliers and used without further purification. The solvents used (dichloromethane and toluene) were dried before use (dichloromethane by distillation over calcium hydride; toluene by distillation over sodium metal with benzophenone as an indicator). The purity of the final compound was determined to be >98%. Nuclear magnetic resonance (NMR) spectra were recorded on an Avance 400 (^1^H, 400 MHz; ^13^C, 100 MHz; ^19^F, 376 MHz) instrument (Bruker, Rheinstetten, Germany) in deuterated chloroform. The chemical shift of the residual protons was used as an internal standard: ^1^H, 7.26 ppm; ^13^C, 77.0 ppm. The chemical shifts of ^19^F peaks were calculated by the instrument without an additional standard. Chemical shifts (δ) are expressed in parts per million (ppm); Coupling constants (J) are given in Hertz (Hz). The reactions were monitored by thin-layer chromatography (TLC) on aluminum sheets with silica gel 60 F254 (Merck, Darmstadt, Germany). Electron ionization mass spectra (EI-MS) and high-resolution mass spectra (HRMS) were recorded on a MAT 95 system (Finnigan, now Thermo Fisher, Waltham, MA, USA) and infrared spectra (IR) on a Bruker Alpha P instrument.

#### 4.1.1. Synthesis of 7-(1-Butylcyclohexyl)-3-(4-fluorobenzyl)-5-methoxy-2H-chromen-2-one 

Under an argon atmosphere, a microwave vial was charged with 300 mg of 4-(1-butylcyclohexyl)-2-hydroxy-6-methoxybenzaldehyde (Scheme 1, compound **1**, 3 mmol, 1.00 equiv.), 314 mg of potassium carbonate (2.27 mmol, 2.20 equiv.), 388 mg of (*E*)-3-(4-fluorophenyl)acrylaldehyde (Scheme 1, compound **2**, 2.58 mmol, 2.50 equiv.) and 275 mg of 1,3 dimethylimidazolium dimethyl phosphate (Scheme 1, compound **3**) (1.24 mmol, 1.20 equiv.) suspended in 4 mL of toluene. The reaction mixture was heated at 110 °C for 50 min via microwave irradiation (Discover LabMate, Mate (CEM, Matthews, NC, USA)). After cooling to room temperature, 10 mL of water was added and the mixture was extracted with 3 × 15 mL of ethyl acetate. Removal of the volatiles under reduced pressure and purification via flash column chromatography (cyclohexanes/ethyl acetate 20:1) resulted in 108 mg (25%) of the pure product as an orange oil. Rf (CH/EtOAc 20:1): 0.25. ^1^H-NMR (CDCl_3_): δ = 7.73 (s, 1 H, C4-H), 7.25–7.29 (m, 2 H, C2‘-HAr, C6‘-HAr), 6.98–7.02 (m, 2 H, C3‘-HAr, C5‘-HAr), 6.88 (s, 1 H, C8-HAr), 6.64 (s, 1 H, C6-HAr), 3.88 (s, 3 H, OCH_3_), 3.84 (s, 2 H, Bn-CH_2_), 1.99–2.04 (m, 2 H, CH_2_C_3_H_7_), 1.25–1.61 (m, 10 H, 5 × CH_2_), 1.08–1.17 (m, 2 H, CH_2_C_2_H_5_), 0.840.92 (m, 2 H, CH_2_CH_3_), 0.76 (t, 3 H, CH_3_, ^3^*J*_HH_ = 7.3 Hz) ppm. ^13^C-NMR (CDCL_3_): δ = 162.9 (Cq, C4‘)162.1 (Cq, C2), 155.3 (Cq, C5), 154.2 (Cq, C8a), 152.9 (Cq, C7), 134.6 (+, C4-H), 134.1 (Cq, C1‘),130.6 (+, C6‘-H), 130.5 (+, C2‘-H), 126.0 (Cq, C3), 115.5 (+, C5‘-H), 115.3 (+, C3‘-H), 107.8 (+, C8-H), 107.6 (Cq, C4a), 103.8 (+, C6-H), 55.7 (+, OCH_3_), 55.7 (Cq, C7-C), 42.2 (−, C3-CH_2_), 36.3 (−, 2 × CH_2_), 36.1 (−, CH_2_-C_3_H_7_), 26.4 (−, CH_2_-C_2_H_5_), 25.6 (−, Cyclohexyl-CH_2_), 23.2 (−, CH_2_CH_3_), 22.4 (−, 2 × Cyclohexyl-CH_2_), 14.0 (+, CH_3_) ppm. ^19^F-NMR (CDCl_3_): δ = –116.9 (sept, C4‘-F) ppm. IR (ATR): = 2928 (m), 2856 (w), 1715 (s), 1613 (s), 1568 (w), 1507 (m), 1453 (m), 1415 (m), 1290 (w), 1252 (m), 1221 (m), 1156 (m), 1112 (s), 1047 (m), 913 (w), 842 (m), 793 (m), 728 (m), 685 (w), 578 (w) cm^−1^. MS (EI): *m*/*z* (%) = 422 (1) [M]^+^, 234 (100) [C_15_H_23_O]^+^, 109 (17) [C_7_H_6_F]^+^. EI-HRMS (C_27_H_31_FO_3_): calc. 422.2257, found. 422.2258 (Scheme 1, Compound **4**).

#### 4.1.2. 7-(1-Butylcyclohexyl)-3-(4-fluorobenzyl)-5-hydroxy-2*H*-chromen-2-one (**KIT 10**)

Under an argon atmosphere, a solution of 4-(1-7-(1-butylcyclohexyl)-3-(4-fluorobenzyl)-5-methoxy-2*H*-chromen-2-one (Scheme 1, compound **4**, 140 mg, 0.330 mmol, 1.0 eq.) in 18 mL of abs. CH_2_Cl_2_ was cooled to −78 °C and BBr_3_ (1 M in CH_2_Cl_2_, 1.66 mL, 1.66 mmol, 5.0 eq.) was added. The solution was stirred for 30 min at that temperature, allowed to warm to room temperature and stirred for an additional 24 h. The reaction was quenched with NaHCO_3_-solution, extracted with 3 × 20 mL CH_2_Cl_2_ and the combined organic layers washed with 30 mL water and 30 mL brine and subsequently dried over Na_2_SO_4_. The volatiles were removed under reduced pressure and the crude product purified via column chromatography (10:1) and the product was received as 117 mg (87%) of a yellow oil. Rf (CH/EtOAc 10:1): 0.19. ^1^H-NMR (CDCl_3_): δ = 7.77 (s, 1 H, C4-H), 7.25–7.28 (m, 2 H, C2‘-HAr, C6‘-HAr), 6.99 (t, 2 H, C3‘-HAr, C5‘-HAr, 3JHH = 8.7 Hz), 6.67 (d, 1 H, C8-HAr, 4JHH = 1.4 Hz), 6.20 (s, 1 H, C6-HAr), 3.85 (s, 2 H, Bn-CH_2_), 1.96–1.98 (m, 2 H, CH_2_-C_3_H7), 1.25–1.54 (m, 11 H, 5 × CH2, OH), 1.06–1.15 (m, 2 H, CH_2_C_2_H5), 0.83–0.90 (m, 2 H, CH_2_CH_3_), 0.75 (t, 3 H, CH_3_, ^3^*J*_HH_ = 7.3 Hz) ppm. ^13^C-NMR (CDCl_3_): δ = 161.7 (Cq, C4‘), 162.6 (Cq, C2), 154.2 (Cq, C5), 153.1 (Cq, C8a), 152.1 (Cq, C7), 134.9 (+, C4-H), 133.9 (Cq, C1‘), 130.6 (+, C6‘-H), 130.6 (+, C2‘-H), 125.8 (Cq, C3), 115.5 (+, C5‘-H), 115.3 (+, C3‘-H), 108.9 (+, C6-H), 107.5 (+, C8-H), 106.7 (Cq, C4a), 41.9 (Cq, C7-C), 36.2 (−, C3-CH2), 36.0 (−, CH2-C3H7), 26.9 (−, 2 × Cyclohexyl-CH_2_), 25.6 (−, Cyclohexyl-CH_2_), 26.4 (−, CH2-C_2_H5), 23.2 (−, CH_2_CH_3_), 22.3 (−, 2 × Cyclohexyl-CH_2_), 14.0 (+, CH3) ppm. ^19^F-NMR (CDCl_3_): δ = –116.4 (sept, C4‘-F) ppm. IR (ATR): = 3293 (w), 2927 (m), 2856 (m), 1678 (s), 1615 (s), 1507 (m), 1452 (m), 1421 (s), 1341 (w), 1275 (m), 1222 (m), 1156 (m), 1065 (m), 907 (m), 847 (m), 795 (m), 729 (s), 669 (w) cm^−1^. MS (EI): *m*/*z* (%) = 408 (7) [M]^+^, 276 (40) [C_17_H_24_O_3_]^+^, 220 (100) [C_13_H_15_O_3_]^+^. − EI-HRMS (C_26_H_29_FO_3_): calc. 408.2101, found. 408.2102.

### 4.2. Biological Part

#### 4.2.1. Ethics Statement

Sprague–Dawley rats were obtained from the Center for Experimental Models and Transgenic Services-Freiburg (CEMT-FR). All the experiments were approved and conducted according to the guidelines of the ethics committee of the University of Freiburg under protocol Nr. X-13/06A (approved on 15 July 2013) and the study were carefully planned to minimize the number of animals used and their suffering.

#### 4.2.2. Chemicals

LPS from *Salmonella typhimurium* (Sigma Aldrich, Deissenhofen, Germany) was resuspended in sterile phosphate-buffered saline (PBS, 5 mg/mL) as stock. Based on our previous studies [8,49,50], the final concentration of LPS used in the cultures was 10 ng/mL. **KIT 10** was dissolved in DMSO.

#### 4.2.3. Primary Rat Microglia Cultures 

As described in our previous studies [49,50], primary mixed glial cell cultures were prepared from cerebral cortices of 1-day neonatal Sprague–Dawley rats. Under sterile conditions, brains were carefully taken, and the cerebral cortices were isolated and the meninges removed. Then, the cortices were gently dissociated and filtered through a 70 μm nylon cell strainer (BD Biosciences, Heidelberg, Germany). After centrifugation at 1000× *g* for 10 min, cells were collected and resuspended in Dulbecco’s modified Eagle’s medium (DMEM from Gibco Life Technologies, Darmstadt, Germany) containing 10% fetal calf serum (Biochrom AG, Berlin, Germany) and antibiotics (40 U/mL penicillin and 40 μg/mL streptomycin, both from Sigma-Aldrich GmbH, Taufkirchen, Germany). Cells were cultured on 10 cm cell culture dishes (Falcon, Heidelberg, Germany) with a density of 5 × 10^5^ cells/mL in 10% CO_2_ at 37 °C (Heracell 240i, Thermo Scientific, distributed by Omnilab, Munich, Germany). With this protocol, it is possible to isolate glial cells (microglia and astrocytes), in which microglia grow on top of the astrocyte layer. These mixed cell cultures reach confluence approximately 2 weeks post-seeding. After 12 days in vitro, microglial cells, which can be seen floating in the medium, were harvested by gentle physical shaking and re-seeded into cell culture plates according to the experimental setup. On the next day, the medium was changed to remove non-adherent cells and after 1 h, the cells were stimulated for respective experiments.

#### 4.2.4. Cell Viability Assay

Viability of primary rat microglia after treatment with **KIT 10** was measured by the CellTiter-Glo^®^ Luminescent Cell Viability Assay (Promega, Mannheim, Germany), which is used to determine the number of metabolically active and viable cells in cell culture based on quantitation of the ATP present in the cells. Briefly, cells were cultured in 96 well plates at the density of 25 × 10^3^ cells/well for 24 h. Then, the medium was changed and after at least 1 h, the cells were incubated with **KIT 10** for 24 h. Since this compound was dissolved in DMSO, the solvent was used in the negative control wells at a final concentration of 0.10% and as a positive control in a higher concentration (10%). The concentration of ATP was measured after 24 h of incubation by adding 100 μL of a reconstituted substrate and incubating for 10 min. Luminescence was measured using a Modulus^TM^ II Microplate Multimode Reader (Turner BioSystems/Promega, Mannheim, Germany).

#### 4.2.5. Determination of PGE_2_ Release from LPS-Activated Microglia 

Cultured primary rat microglia were incubated with **KIT 10** (0.1–25 µM) for 30 min. Afterward, the cells were treated with or without LPS (10 ng/mL) for the next 24 h. Supernatants were harvested and levels of PGE_2_ were measured using a commercially available enzyme immunoassay (EIA) kit (Cayman Chemical, Ann Arbor, MI, USA). Standard concentrations of 39–2500 pg/mL were used and the sensitivity of the assay was 36 pg/mL. The results were normalized to LPS and presented as a percentage of change in PGE_2_ levels of at least three independent experiments.

#### 4.2.6. Immunoblotting

Rat primary microglia were treated with **KIT 10** and controls (0.1–10 µM) for 30 min; then, LPS (10 ng/mL) was added for different time points (depending on the studied protein). After the experiment, cells were washed with cold PBS and lysed in lysis buffer (42 mM Tris–HCl, 1.3% sodium dodecyl sulfate, 6.5% glycerin, 100 μM sodium orthovanadate, and 2% phosphatase and protease inhibitors). The protein concentration of the samples was measured using the bicinchoninic acid (BCA) protein assay kit (Fisher Scientific GmbH, Schwerte, Germany) according to the manufacturer’s instructions. For western blotting, 15 μg of total protein from each sample was subjected to sodium dodecyl sulfate-polyacrylamide gel electrophoresis (SDS-PAGE) under reducing conditions. Afterward, proteins were transferred onto polyvinylidene fluoride (PVDF) membranes (Merck Millipore, Darmstadt, Germany) by semi-dry blotting. After blocking with Roti-Block (Roth, Karlsruhe, Germany), membranes were incubated overnight with primary antibodies. Primary antibodies were goat anti-COX-2 (1:500; Santa Cruz Biotechnology, Heidelberg, Germany), rabbit anti-mPGES-1 (1:6000; Agrisera, Vännas, Sweden), mouse anti-inhibitor of kB (IkB)-α (1:500; Santa Cruz Biotechnology), rabbit anti-ERK 1/2 (1:1000; Cell Signaling Technology, Frankfurt, Germany), rabbit anti-p38 MAPK (1:1000; Cell Signaling Technology), rabbit anti-JNK (1:1000; Cell Signaling Technology), and rabbit anti-actin (1:5000; Sigma-Aldrich). The proteins were detected with horseradish peroxidase-coupled rabbit anti-goat IgG (Santa Cruz, 1:100,000 dilution), goat anti-rabbit IgG (Amersham, 1:25,000 dilution), or mouse anti-rabbit IgG (Amersham, 1:25,000 dilution) using enhanced chemiluminescence (ECL) reagents (GE Healthcare, Freiburg, Germany). Densitometric analysis was performed using ImageJ software (NIH, Bethesda, MD, USA) and β-actin control was used to confirm equal sample loading and normalization of the data.

#### 4.2.7. Cyclooxygenases Activities Assay in Primary Microglia Culture

The COX enzymes are responsible for the formation of prostanoids through arachidonic acid (AA). Under an inflammatory stimulus, the enzyme COX-2 is one of the enzymes responsible to synthesize prostaglandins [29]. Under unstimulated conditions, primary microglial cells only express the COX-1 isoform [37]. 

To measure COX-1 activity, primary rat microglial cells were plated in 24-well cell culture plates. After 24 h, the medium was removed and replaced with a serum-free medium. **KIT 10** (0.1–10 µM) or the selective COX-1 inhibitor SC560 (0.1–10 µM, reversible inhibitor) were added and left for 15 min. Then, 15 µM of AA (substrate) was supplemented for another 15 min. Supernatants were then collected and used for the determination of PGE_2_. 

To measure COX-2 activity, primary rat microglial cells were plated in 24-well cell culture plates, and pre-incubated with LPS (10 ng/mL) for 24 h. Then, the medium was removed, and replaced with a serum-free medium. **KIT 10** (0.1–10 μM) or diclofenac sodium (non-selective COX inhibitor, 10 µM) were added and left for 15 min. Then, 15 μM of AA (substrate) was added for another 15 min. Supernatants were then collected and used for the determination of PGE_2_.

#### 4.2.8. Statistical Analysis

Statistical analyses were performed using Prism 5 software (GraphPad Software Inc., San Diego, CA, USA). Values of all experiments were represented as mean ± SEM of at least three independent experiments. Raw values were converted to a percentage and positive control was considered as 100%. Values were compared using one-way ANOVA with post hoc Student-Newman-Keuls test (multiple comparisons). The level of significance was set at * *p* < 0.05, ** *p* < 0.01, and *** *p* < 0.001. IC_50_ was calculated using Prism 5 software following the nonlinear regression, “Dose-Response-Inhibition”.

## Data Availability

The data presented in this manuscript are available from the corresponding author upon request.

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
