# Peer review of "Effects of a Novel GPR55 Antagonist on the Arachidonic Acid Cascade in LPS-Activated Primary Microglial Cells"

_ijms, 2021, doi:10.3390/ijms22052503_

Round 1

Reviewer 1 Report

Dear Authors,

an interesting manuscript on a GPR.

There are many minor grammar mistakes throughout the manuscript.  Thus, it is needed that a native speaker reads through the manuscript to correct them.

In the abstract it is needed that the authors give the concentrations of LPS and KIT10 applied in this study.

The sentence from line 25 to 28 is an overstatement.  The results are not conclusive in this respect.   It can be said that a correlation is found.  In particular take care should be taken in mentioning  COX-2 in this way in the abstract, because no effects on  COX-2 activity were found.  

The effects on IκB (in relation to NF-κB) should be mentioned in the abstract i.e. IκB should be mentioned.

In the Introduction, please indicate which extra information obtained with/for KIT10 was expected, in comparison to the extant antagonist CID16020046.

line 68 intracellular ( instead of intercellular )

line 77 cannabinoids ( instead of cannabinoids ligands )

line 88-89 associated with ( instead of dependent on )

Regarding  Figure 1.  What is indicated by "3." is not clear .  For example, the fragment " yields coumarin 3 with 25% " in line 100 appears incorrect and incomplete.

In Figure 2. Instead of application of DMSO  indicated with "10" indicate it with "+" (like for LPS)

Regarding the application of Actin as a loading control in this study.  In figures 4, it appears that actin expression is reduced by increasing KIT10 concentrations.   By looking at the ratio between the expression of the protein of interest and the actin expression, the reduction of the expression of the protein of interest is obscured.  But maybe the "uneven" staining  for both KIT10 and the proteins of interest is a staining artefact.

Effects on actin expression by various treatments is well known.  Thus, differences between groups very well can be "true" differences.  Actin is first of all good to recognize intra group sample variability (not inter group).  It could be useful to assay other loading controls that could be more useful for intergroup comparisons.  I would say that if intragroup variability is small, one can maybe rely on the accuracy of the sample amounts. 

Okay, in figure 6, higher concentration of KIT10 do not appear to lead to reduction in Actin expression, but rather an increase.   Thus, the authors choice to use Actin to compare between groups maybe a good one.

It is a surprise that no effects on COX-1 and COX-2 activity are found (given the reduction in COX-2 expression) .  It begs the question whether it would not have been prudent to have 25 µM KIT10 applied as well.  This concentration might have brought about an effect that is not seen with the lower concentrations.

Also a question, Why was 10ng/mL chosen for LPS ?  Maybe a lower concentrations could have allowed for clearer effects of KIT10.  Was a dose response curve for LPS effects prepared ?

Regarding the Discussion, the effects on IκB (in relation to NF-κB) should be given more discussion.  For example, discuss what is the mechanism whereby GPR55 can have an effect on IκB .

In general, how can GPR55 have an effect on the expression of the proteins studied.  Is this known ?

The sentence running from 201 to 206 should be broken up into 2 or 3 sentences.

In line 277 What is 4 ?   coumarin one can suspect.

Instead of describing centrifugation by "rpm" on should describe it with "×g"

350-353, describe clearer when floating or adherent cells are taken, and the reasons.

The Methods provide a good place to explain why the concentration of 10ng/mL for LPS was  chosen .

10-20 µg for Western blotting.   Say for which figures which amount was used

Say why arachidonic acid was applied.  Yes, I am intelligent, I can figure it.  But it is better when it is just written down where the knowledge is needed.

In the author contributions say "KIT10 the GPR55 antagonist of this study."

Author Response

Reviewer 1

Dear Authors,

an interesting manuscript on a GPR.

There are many minor grammar mistakes throughout the manuscript.  Thus, it is needed that a native speaker reads through the manuscript to correct them.

Response: We appreciate the reviewers comment and revised the text accordingly.

In the abstract it is needed that the authors give the concentrations of LPS and KIT10 applied in this study.

Response: We have improved the manuscript by adding more details in the abstract (lines 27 - 29).

The sentence from line 25 to 28 is an overstatement.  The results are not conclusive in this respect.   It can be said that a correlation is found.  In particular take care should be taken in mentioning COX-2 in this way in the abstract, because no effects on COX-2 activity were found.  

Response: We agree with the reviewer and revised the abstract accordingly (please, see lines 20 - 34).

The effects on IκB (in relation to NF-κB) should be mentioned in the abstract i.e. IκB should be mentioned.

Response: We have included IκB in the abstract (please, see lines 31 - 33).

In the Introduction, please indicate which extra information obtained with/for KIT10 was expected, in comparison to the extant antagonist CID16020046.

Response: Thanks for this comment. Honestly, we didn’t have any expectations since KIT10 and CID16020046 have a complete structure. Further studies using this commercial antagonist are needed to elucidate differences in their action.

line 68 intracellular ( instead of intercellular )

Response: We revised this point in the text (line 75).

line 77 cannabinoids ( instead of cannabinoids ligands )

Response: We revised this point in the text (line 84).

line 88-89 associated with ( instead of dependent on )

Response: We revised this point in the text (line 95).

Regarding  Figure 1.  What is indicated by "3." is not clear .  For example, the fragment " yields coumarin 3 with 25% " in line 100 appears incorrect and incomplete.

Response: Thanks for the good comment, we tried to write this part clearer (lines 105 - 108).

Coumarin 3 (Fig.1) is an Imididazolimphosphate which is derivatized in a subsequent reaction step leading to coumarin 4 (Fig.1), where a deprotection with boronic tribromide lead to CCCCC1(CCCCC1)c1cc(O)c2c(c1)oc(=O)c(c2)Cc1ccc(cc1)F (named KIT 10). We revised this part.

In Figure 2. Instead of application of DMSO  indicated with "10" indicate it with "+" (like for LPS)

Response: We revised this point in the text (line 124).

Regarding the application of Actin as a loading control in this study. In figures 4, it appears that actin expression is reduced by increasing KIT10 concentrations. By looking at the ratio between the expression of the protein of interest and the actin expression, the reduction of the expression of the protein of interest is obscured.  But maybe the "uneven" staining for both KIT10 and the proteins of interest is a staining artefact.

Effects on actin expression by various treatments is well known. Thus, differences between groups very well can be "true" differences. Actin is first of all good to recognize intra group sample variability (not inter group). It could be useful to assay other loading controls that could be more useful for intergroup comparisons.  I would say that if intragroup variability is small, one can maybe rely on the accuracy of the sample amounts. 

Okay, in figure 6, higher concentration of KIT10 do not appear to lead to reduction in Actin expression, but rather an increase.   Thus, the authors choice to use Actin to compare between groups maybe a good one.

Response: We appreciate your comment and agree that the loading control needs to be carefully chosen to avoid any false / positive results and as outlined by the reviewers there might be staining and blotting errors.

It is a surprise that no effects on COX-1 and COX-2 activity are found (given the reduction in COX-2 expression). It begs the question whether it would not have been prudent to have 25 µM KIT10 applied as well.  This concentration might have brought about an effect that is not seen with the lower concentrations.

Response: We agree with the reviewer that it would be of interest to investigate the effects of KIT10 at 25 µM. However, we believe that it will not change what was observed for the 10 µM, since we don´t see any concentration depending tendency to reduce on COX enzyme  activities, moreover, the inhibition of PGE2 using 10 µM (81 %) was nearly the same as the 25 µM (75 %) (Fig. 3).

Moreover, our group has demonstrated that different compounds interfere with the activity of the enzymes, but do not affect the expression and synthesis of COX and vice-versa (such as Candelario-Jalil et al., J Neuroinflammation. 2007; 4: 25. doi: 10.1186/1742-2094-4-25; Bhatia et al., Sci Rep. 2017; 7: 116. doi: 10.1038/s41598-017-00225-5; Gargouri et al., Phytomedicine. 2018 May 15;44:45-55. doi: 10.1016/j.phymed.2018.04.009; Saliba et al., J Neuroinflammation. 2018; 15: 322. doi: 10.1186/s12974-018-1362-7). We have demonstrated in a previous study (Saliba et al., 2018) that another coumarin derived synthetic compound (KIT17) strongly inhibited LPS-induced PGE2 release and reduced mPGES-1 and COX-2 protein synthesis, but had no effects on COX enzyme activity. We believe that more studies are necessary to further understand the diverse and complex pharmacology of GPR55 and its role in the arachidonic acid pathway.

Also a question, Why was 10ng/mL chosen for LPS? Maybe a lower concentrations could have allowed for clearer effects of KIT10. Was a dose response curve for LPS effects prepared?

Response: We appreciate your good comment. Our group has been studying the effects of LPS in microglia for years and we established the best concentration of LPS to use for this cell type and the use of anti-inflammatory drugs based on various previous results (e.g. de Oliveira et al., Glia . 2008 Jun;56(8):844-55. doi: 10.1002/glia.20658).

Regarding the Discussion, the effects on IκB (in relation to NF-κB) should be given more discussion.  For example, discuss what is the mechanism whereby GPR55 can have an effect on IκB.

Response: This is a good point and a few suggestions regarding the upstream mechanisms are included in the discussion part (lines 259 – 264).

Further studies have to be performed to elucidate the upstream step on which KIT10 exactly prevents the degradation of Iκ. Besides interfering with the activation of IL-1R-associated kinase (IRAK) 1/2 and IKK, GPR55 antagonists might directly act at recruited adapters of TLR4 such as myeloid differentiation factor (MyD) 88 and TNF-receptor-associated factor (TRAF6).

In general, how can GPR55 have an effect on the expression of the proteins studied. Is this known?

Response: This is a very good question, we have discussed quite often in the group. We do not know yet by which intracellular signaling molecules GPR55 affects the described pathways and protein synthesis. Future studies, especially on the receptor itself and adapters, are needed to bring more light in the aspect.

The sentence running from 201 to 206 should be broken up into 2 or 3 sentences.

Response: We revised this point in the text (line 203 - 216).

In line 277 What is 4? coumarin one can suspect.

Response: The number 4 refers to the coumarin 4 described on the figure 1. We revised this point in the text (line 300 – 305, 328).

Instead of describing centrifugation by "rpm" on should describe it with "×g"

Response: We revised this point in the text (line 369).

350-353, describe clearer when floating or adherent cells are taken, and the reasons.

Response: We have improved the manuscript by adding more details in the material and methods section on lines 374 - 380.

The Methods provide a good place to explain why the concentration of 10ng/mL for LPS was chosen.

Response: We included our previous studies in the material and methods sections (lines 360-362).

10-20 µg for Western blotting.   Say for which figures which amount was used

Response: Thank you for this observation. All experiments were done using 15 µg of total protein of the cell lysates. Sorry, the other information was wrong, 10 and 20 µg/ml were used for other Western blots not shown in the manuscript. We revised this point in the text (line 412).

Say why arachidonic acid was applied.  Yes, I am intelligent, I can figure it.  But it is better when it is just written down where the knowledge is needed.

Response: The COX enzymes are responsible for the formation of prostanoids using arachidonic acid (AA) as substrate. Thus, in this protocol, AA is used as substrate for the cellular COX enzymes to generate PGs in minutes (lines 429, 430, 436, 443). In contrast, COX-2 mediated PGE2 release occurs at a later time point since the protein first needs to be synthesized.

In the author contributions say "KIT10 the GPR55 antagonist of this study."

Response: We added KIT 10 to the sentence (line 454-455).

Reviewer 2 Report

Overall: This is a straightforward paper looking at the effects of a new GPR55 antagonist (KIT10) on microglial proinflammatory signaling. The data are sound and the conclusions straight forward. This report is similar to a previously published paper by the group in 2018 using a related compound (KIT17, PMID: 30453998). Therefore, it is important for the authors to convey why this antagonist is different and/or better than KIT17.  Otherwise, this is a nice brief report of the anti-inflammatory potential of a new GPR55 antagonist.

Introduction:

  • Line 52 please provide the references for GPR55 expression in microglia
  • Line 51 paragraph might be better introduced by stating that the roles of GPR55 in the CNS and PNS might differ. It is a little difficult to tell by reading this section of introduction if the GPR55 antagonist was expected to blunt microglial proinflammatory responses or promote them.
  • Please clarify if GPR55 is a Gi, Gq, or Gs coupled GPCR
  • Please provide background as to how KIT17 differs from KIT10, and why it is a different or better compound to bring out the novelty of this report.

Results:

  • Figure 3 nicely shows KIT blunts microglial PGE2 production
  • The explanation in section 2.2.4 is not clear. I think the interpretation is KIT had no effect on COX activity at baseline. However, in the setting of inflammatory activation by LPS, KIT blunted PGE2 production, showing its mechanism of action is likely not due to direct inhibition of COX.

Author Response

Reviewer 2:

Overall: This is a straightforward paper looking at the effects of a new GPR55 antagonist (KIT10) on microglial proinflammatory signaling. The data are sound and the conclusions straight forward. This report is similar to a previously published paper by the group in 2018 using a related compound (KIT17, PMID: 30453998). Therefore, it is important for the authors to convey why this antagonist is different and/or better than KIT17.  Otherwise, this is a nice brief report of the anti-inflammatory potential of a new GPR55 antagonist.

Introduction:

  • Line 52 please provide the references for GPR55 expression in microglia

Response: We revised this point in the text (line 60).

Line 51 paragraph might be better introduced by stating that the roles of GPR55 in the CNS and PNS might differ. It is a little difficult to tell by reading this section of introduction if the GPR55 antagonist was expected to blunt microglial proinflammatory responses or promote them.

Response: Thanks a lot for this good comment. The pharmacology of GPR55 remains controversial and depends on the cell type and stimulation agent (line 61,62). For a better understanding of the role of GPR55 in the context of neuroinflammation, more studies are needed.

Please clarify if GPR55 is a Gi, Gq, or Gs coupled GPCR

Response: We revised this point in the text (line 73).

  • Please provide background as to how KIT17 differs from KIT10, and why it is a different or better compound to bring out the novelty of this report.

Response: Thanks a lot for this good comment.

KIT-17 (3-benzyl-6-hydroxy-5,7,8-trimethyl-2H-chromen-2-one) is a small generic coumarin which is structurally different from KIT-10 (7-(1-butylcyclohexyl)-3-(4-fluorobenzyl)-5-hydroxy-2H-chromen-2-one). The latter is inspired by tetrahydrocoumarin (THC). The position of the hydroxy groups (meta/para) causes a different acidity and thus different binding to proteins. Also, the bulky cyclohexyl ring in KIT-10 increased the hydrophilic character of this compound in the western portion. All in all, KIT 10 and KIT17 are very different compounds (line 277 - 284).

Results:

  • Figure 3 nicely shows KIT blunts microglial PGE2 production

Response: We appreciate your comment.

  • The explanation in section 2.2.4 is not clear. I think the interpretation is KIT had no effect on COX activity at baseline. However, in the setting of inflammatory activation by LPS, KIT blunted PGE2 production, showing its mechanism of action is likely not due to direct inhibition of COX.

Response: We agree with the reviewer and revised this point in the text (line 156).
